# Intrauterine growth restriction and its associated factors in Tehran, comparing 3 common standards

Mahtab Toulany[1], Narjes Khalili[1], Mohammad Heidarzadeh[2], Abbas Habibelahi[3], Arghavan Haj-sheykholeslami [1]*

1 Preventive Medicine and Public Health Research Centre, Psychosocial Health Research Institute, Department of Community and Family Medicine, School of Medicine, Iran University of Medical Sciences, Tehran, Iran, 2 Department of Pediatrics, Tabriz University of Medical Sciences, Tabriz, Iran, 3 Neonatal Health Office, Ministry of Health and Medical Education, Tehran, Iran

* arghavan_sheykh@yahoo.com, hajsheykholeslami.a@iums.ac.ir

## Abstract

### Objectives

Several standard charts have been proposed for the diagnosis of intrauterine growth restriction (IUGR) at birth but no global or national consensus exists on using any of them. We aimed to evaluate and compare the prevalence of IUGR in Tehran using 3 common standards and identify the associated factors.

### Methods

Using the Iranian Maternal and Neonatal Network registry, we extracted the data of all singleton live births in Tehran province of Iran in 2018 to eliminate the possible confounding effects of the COVID-19 pandemic. We defined IUGR as having a birth weight less than the 10th percentile for gestational age using 3 standards including World Health Organization's and INTERGROWTH-21st charts and the same population's 10th percentiles. Logistic regression was used to identify the associated factors.

### Results

There were 187031 singleton live births. The prevalence of IUGR using WHO, INTERGROWTH-21st, and the population's 10th percentile was 11.8, 4.2, and 9.7 percent respectively; Among these, 7681 cases (4.1%) were identified by all 3. Neonatal trisomy 21, maternal addiction, eclampsia/pre-eclampsia, chronic hypertension, history of abortion, Primigravidity, being older than 35 yrs. and parental consanguinity were positively associated with IUGR where mother's gestational diabetes, higher education level, delivering the baby in a private hospital and living in Paakdasht or Shahryar cities were inversely associated with IUGR.

**Data availability statement:** The data underlying the results presented in the study are owned by and available from Iran's Ministry of Health and Medical Education. The data to be asked for is the singleton livebirths in Tehran province in 2018 from the IMAN registry. The way to communicate with the Ministry is via the Contact Us form on the ministry's website at https://behdasht.gov.ir.

**Funding:** The author(s) received no specific funding for this work.

**Competing interests:** The authors have declared that no competing interests exist.

## Conclusion

The IUGR prevalence highly depended on the standards used ranging from 4.2 to 11.8%, showing a great need for a global consensus. Neonatal trisomy 21, maternal addiction and eclampsia/pre-eclampsia had the strongest positive associations with IUGR.

## Background

Intrauterine growth restriction (IUGR) is a condition in which the fetus's growth in the uterus is slower than normal [1–3]. Globally IUGR is diagnosed in about 10–15% of all births with 75% of the cases occurring in developing countries [4]. After prematurity, IUGR is the second leading cause of perinatal mortality [5]. It is associated with stillbirth, neonatal mortality, perinatal complications and even increased risk of other morbidities such as metabolic syndrome, diabetes, hypertension, cardiovascular disease, mental illness and social problems in the future [6–9].

A wide variety of conditions, including embryonic, placental, and maternal factors, predispose a fetus to IUGR. However, maternal factors are the most predominant causes [4,10]. There are multiple standards for at birth IUGR diagnosis but there's no global consensus on which should be used. Limited studies have covered this important public health issue in our community and no national standard has been specified.

In this study, we aimed to evaluate the prevalence of IUGR and its associated factors in Tehran province which has more than 13 million residents accounting for 16.6% of Iran's total population.

## Methods

This cross-sectional study was conducted on neonates born in Tehran province of Iran. To omit the COVID-19 pandemic influence, we chose the data from 2018, the year exactly before the start of the epidemic in Iran. In collaboration with the Neonatal Health Office of the health ministry, the anonymized demographic and preliminary data were extracted from the Iranian Maternal and Neonatal Network (IMaN), which is a national network that registers the birth information of all in and out of hospital deliveries throughout the country (Approximate date of data acquisition 20-12-2021).

All singleton live births with a gestational age of at least 26 weeks in Tehran during 2018 were included in the study in a census manner. Using a checklist, we extracted information related to these births including demographic and health information of mothers and their neonates. All records were carefully reviewed to exclude those with missing data.

The definition we used for IUGR diagnosis was having a birth weight less than the10th percentile for gestational age using 3 different most popular standards including the World Health Organization's fetal growth charts [11], INTERGROWTH-21st standards [12] and the same population's 10th percentiles [10]. To obtain Tehran newborns' birth weight percentiles, we categorized our data (information on all

singleton live births in Tehran in 2018) by weeks of gestation and calculated the 10th percentiles for each gestational age in our population. The gestational age was calculated using the first day of the last period and the first-trimester ultrasound results in mothers with regular and irregular menstrual cycles, respectively. Based on gestational age, the neonates were categorized into three major groups of pre-term (from 26 to less than 37 weeks), term (from 37 to 41 weeks), and post-term (42 weeks and above) for univariate analysis. For multivariate analysis, only those recognized by all 3 standards were considered as IUGR cases.

## Statistical analysis

All analyses were carried out using SPSS v.16 software (Build 1.0.0.1347; IBM, New York, USA.). The Chi-square test was used for univariate comparison of those with and without IUGR. The variables with less than 0.2 p values in the univariate analysis, were included into the logistic regression model for multivariate analysis, and adjusted odds ratios were subsequently calculated. Since birth weight and gestational age were used in determining the IUGR status of the newborns, these 2 variables were excluded from the model. The delivery type was also excluded because it was highly related to the size of the neonate and did not have a role in the intrauterine growth of the fetus. Delivering the baby in a private hospital was kept in the model as a proxy for the parents' socio-economic status. Agreement between the 3 standards was assessed by calculating the Kappa measure of agreement.

## Ethical considerations

This study was approved by the Iran University of Medical Science's ethics review committee (Ethics Code: IR.IUMS. FMD.REC.1398.328). The authors complied with the World Medical Association Declaration of Helsinki regarding ethical conduct of research involving human subjects and/or animals.

## Results

A total number of 195499 births occurred in 2018 in Tehran province. Out of this, 1330 (0.7%) cases were stillbirths and 6764 cases were the results of a non-singleton pregnancy which were excluded. At the end, a total of 187031 cases entered the study. Intrauterine growth restriction was identified in 11.8, 4.2 and 9.7 percent using WHO, INTER GROWTH-21st and the population's 10th percentiles respectively. Among these, 7681 cases (4.1% of all singleton live births) were identified as having IUGR by all of the 3 used standards. Among our 174736 singleton live births with 37 or more weeks of gestation, 3893 (2.2%), had birth weights less than 2500g while 86.6% of them fulfilled the criteria for IUGR diagnosis, 520 cases (13.4%) did not. On the other hand, among the remaining 170843 neonates weighing 2500g or more at birth, 2.1% were also identified as IUGR cases by all the measures used.

The results of univariate analysis, showing the associations between IUGR and demographic, maternal and neonatal factors are shown in Table 1. Prevalence of IUGR was significantly higher in neonates with trisomy 21, those born to mothers with eclampsia/ preeclampsia, chronic hypertension, addiction, history of abortion, older than 35 years of age and those experiencing their first pregnancy. Intrauterine growth restriction was also more prevalent in post-term babies, those delivered via vaginal delivery or those born in non-private hospitals. Parental consanguinity also showed a significant association with IUGR.

Table 2 summarizes the results of the multivariate logistic regression analysis. Our final results showed that the neonate's trisomy 21 and mother's addiction, eclampsia/ pre-eclampsia, chronic hypertension, history of abortion, primigravidity, being older than 35 years and parental consanguinity were positively associated with IUGR occurrence where the mother's gestational diabetes, higher levels of education, delivering the baby in a private hospital and living in Paakdasht or Shahryar cities showed negative associations. Best agreement was observed between Tehran neonates' 10th percentiles with WHO standards with a Kappa of 0.79 where the kappa between this local standard and INTERGROWTH-21 charts was 0.56 while it was 0.48 between the WHO and the INTERGROWTH-21 standards.

**Table 1. Association between intrauterine growth restriction and demographic, maternal and neonatal factors in univariable analysis.**

| Variable | Number of singleton live births in each category (%) | Intrauterine Growth Restriction (IUGR) frequency | P value |
|---|---|---|---|
| Mother's level of education | High school not finished = 46198(24.7) | 4.6% | P<0.001 |
| | High school diploma = 68433 (36.6) | 4.1% | |
| | University degree = 70144 (37.5) | 3.8% | |
| | Missing = 2256 (1.2) | | |
| Mother's previous IUFD* | Yes = 1530 (0.8) | 4.8% | 0.15 |
| | No = 185501(99.2) | 4.1% | |
| Neonate's sex | Male = 96992(51.9) | 4.1% | 0.34 |
| | Female = 90039 (48.1) | 4.2% | |
| Delivery type | Vaginal delivery = 59768 (32) | 4.6% | P<0.001 |
| | Caesarean section = 127263(68) | 3.9% | |
| Mother's chronic Hypertension | Yes = 3109(1.7) | 7% | P<0.001 |
| | No = 183922 | 4.1% | |
| Mother's cardiac diseases | Yes = 1371(0.7) | 4.2% | 0.92 |
| | No = 185446 | 4.1% | |
| Mother's Autoimmune diseases | Yes = 500(0.3) | 5.4% | 0.14 |
| | No = 186318 | 4.1% | |
| Mother's Diabetes Mellitus | Yes = 849(0.5) | 4.1% | 0.88 |
| | No = 185969 | 4% | |
| Mother's psychologic diseases | Yes = 754(0.4) | 4.5% | 0.57 |
| | No = 186062 | 4.1% | |
| Mother's Hypothyroidism | Yes = 19525(10.5) | 3.9% | 0.14 |
| | No = 167302 | 4.1% | |
| Mother's Gestational Diabetes | Yes = 9034(4.8) | 3% | P<0.001 |
| | No = 177807 | 4.2% | |
| Mother's Eclampsia/Pre-eclampsia | Yes = 2973(1.6) | 9.8% | P<0.001 |
| | No = 183851 | 4% | |
| Mother's Hepatitis B virus infection | Yes = 432(0.2) | 4.2% | 0.95 |
| | No = 186384 | 4.1% | |
| Mother's HIV** infection | Yes = 29(0.01) | 3.4% | 0.85 |
| | No = 186787 | 4.1% | |
| Mother's positive VDRL*** test | Yes = 44(0.02) | 9.1% | 0.1 |
| | No = 186772 | 4.1% | |
| Mother's cigarette smoking habit | Yes = 111(0.1) | 12.6% | P<0.001 |
| | No = 186706 | 4.1% | |
| Mother's addiction | Yes = 303(0.2) | 13.9% | P<0.001 |
| | No = 186513 | 4.1% | |
| Mother's alcohol use | Yes = 5(0.002) | 20% | 0.18 |
| | No = 186816 | 4.1% | |
| Trisomy 21 in the neonate | Yes = 11(0.006) | 18.2% | 0.07 |
| | No = 187020 | 4.1% | |
| City of residence | Tehran = 163597(87.5) | 4.1% | 0.004 |
| | Shahryar = 8890(4.8) | 3.8% | |
| | Varamin = 3364(1.8) | 4.8% | |
| | Eslamshahr = 2801(1.5) | 3.7% | |
| | Paakdasht = 2083(1.1) | 3% | |
| | Robatkarim = 1856(1) | 5.1% | |
| | Shahre Ghods = 1057(0.6) | 4.2% | |
| | Gharchak = 219(0.1) | 3.7% | |
| | Cities < 100,000 residents = 3164(1.7) | 4.7% | |

*(Continued)*

**Table 1.** (Continued)

| Variable | Number of singleton live births in each category (%) | Intrauterine Growth Restriction (IUGR) frequency | P value |
|---|---|---|---|
| Delivery in a private hospital | Yes = 66142(35.4) | 3.7% | P < 0.001 |
| | No = 120889(64.6) | 4.1% | |
| Parental consanguinity | Yes = 22701(12.1) | 4.8% | P < 0.001 |
| | No = 164329(87.9) | 4% | |
| Mother's age | Younger than 18 Yrs. = 1391(0.7) | 6.6% | P < 0.001 |
| | 18 to 35 Yrs. old = 152693(81.6) | 4.1% | |
| | Older than 35 Yrs. = 32947(17.6) | 4% | |
| Mother's first pregnancy | Yes = 70003(37.4) | 5% | P < 0.001 |
| | No = 117028(62.6) | 3.5% | |
| Mother's history of abortion | Yes = 39567(21.2) | 3.8% | P < 0.001 |
| | No = 147464(78.8) | 4.2% | |
| Birth weight | Under 2500 g = 9259(5) | 45.6% | P < 0.001 |
| | 2500 to 4000g = 171981(92) | 2.1% | |
| | More than 4000 g. = 5791(3.1) | 0% | |
| Term delivery | Preterm = 12295 (6.6) | 6.2% | P < 0.001 |
| | Term = 174427 (93.3) | 3.9% | |
| | Post term = 309(0.2) | 1.7% | |

*Intrauterine fetal demise,

**Human Immunodeficiency Virus,

***Venereal Disease Research Laboratory test

## Discussion

This study investigated IUGR prevalence and its associated factors in singleton live births of Tehran province in Iran just before the COVID-19 pandemic comparing the results of using 3 different standards for its diagnosis. We found a prevalence of 11.8, 4.2 and 9.7% using WHO, INTERGROWTH-21st and the population's 10th percentile standards respectively. According to the results, IUGR was more common in neonates with Down syndrome or those born to mothers with addiction, eclampsia/pre-eclampsia, chronic hypertension, history of abortion and also mothers experiencing their first pregnancy or older than 35 years of age. Parental consanguinity was also positively associated with IUGR where the mother's gestational diabetes and higher educational level showed negative associations. Delivering the baby in a private hospital as a socio-economic status proxy and residing in Paakdasht or Shahryar cities were also negatively associated with its occurrence. The highest and lowest prevalence was measured using WHO and INTERGROWTH-21st standards respectively and the best agreement existed between WHO and Tehran newborns' birth weight 10th percentiles with a Kappa of 0.79.

We could not find any Iranian study that used any of our 3 selected standards for the IUGR definition, making it really hard to compare the results. The closest study was conducted in Ilam (a province in the western part of Iran) in 2015, which reported a prevalence of 2.8% using Fenton growth charts for IUGR definition [13], a substantially lower rate than what we found in Tehran in our study. In studies performed in other countries, Zepeda-Monreal et al. reported a prevalence of 13.5% in Mexico in 2009, which is higher than our results [14]. However, in Spain in 2009, Romo et al. found an IUGR prevalence of 5.13% [15], and Liu et al. reported a prevalence of 5.6% in Chinese newborns in 2001 [16], which are closer to our findings. In a more recent Ethiopian study, IUGR prevalence in 2019 was considerably higher than the other results at 23.5% [4]. It should be emphasized that regardless of the different times the studies have been conducted, different inclusion criteria and different socio-economic and developmental states of the countries, using different standards for IUGR diagnosis in each of these studies, makes comparing the results nearly impossible [13,14,17–23], as shown in

**Table 2. Logistic regression results for associations between intrauterine growth restriction and demographic, maternal and neonatal factors.**

| Variables | categories | Exponential β | 95% Confidence Interval | P-value |
|---|---|---|---|---|
| Mother's age | < 18 Yrs. | 1.17 | 0.94,1.45 | 0.17 |
| | 18 to 35Yrs. | Reference | – | – |
| | >35 Yrs. | 1.09 | 1.03,1.17 | 0.006 |
| Mother's level of education | High school not finished | Reference | – | – |
| | High school diploma | 0.87 | 0.82,0.93 | P<0.001 |
| | University degree | 0.80 | 0.75,0.86 | P<0.001 |
| Mother's first pregnancy | No | Reference | – | – |
| | Yes | 1.62 | 1.53,1.71 | P<0.001 |
| Mother's history of abortion | No | Reference | – | – |
| | Yes | 1.12 | 1.05,1.19 | 0.001 |
| Mother's previous IUFD* | No | Reference | – | – |
| | Yes | 1.23 | 0.97,1.57 | 0.09 |
| Mother's chronic Hypertension | No | Reference | – | – |
| | Yes | 1.75 | 1.52,2.02 | P<0.001 |
| Mother's Autoimmune diseases | No | Reference | – | – |
| | Yes | 1.33 | 0.89,1.98 | 0.16 |
| Mother's Hypothyroidism | No | Reference | – | – |
| | Yes | 0.34 | 0.96,0.89 | 0.34 |
| Mother's Gestational Diabetes | No | Reference | – | – |
| | Yes | 0.66 | 0.59,0.75 | P<0.001 |
| Mother's Eclampsia/Pre-eclampsia | No | Reference | – | – |
| | Yes | 2.54 | 2.24,2.88 | P<0.001 |
| Mother's positive VDRL test ** | No | Reference | – | – |
| | Yes | 2.13 | 0.76,5.97 | 0.15 |
| Mother's cigarette smoking habit | No | Reference | – | – |
| | Yes | 1.6 | 0.84,3.05 | 0.15 |
| Mother's addiction | No | Reference | – | – |
| | Yes | 2.99 | 2.07,4.3 | P<0.001 |
| Mother's alcohol use | No | Reference | – | – |
| | Yes | 1.84 | 0.18,19.03 | 0.61 |
| Trisomy21 in the neonate | No | Reference | – | – |
| | Yes | 5.92 | 1.25,28.1 | 0.02 |
| City of residence | Tehran | Reference | – | – |
| | Shahryar | 0.85 | 0.76,0.95 | 0.006 |
| | Varamin | 1.08 | 0.92,1.28 | 0.34 |
| | Eslamshahr | 0.86 | 0.70,1.05 | 0.13 |
| | Paakdasht | 0.63 | 0.48,0.82 | P<0.001 |
| | Robatkarim | 1.13 | 0.91,1.39 | 0.27 |
| | Shahre Ghods | 0.95 | 0.70,1.28 | 0.72 |
| | Gharchak | 0.86 | 0.42,1.73 | 0.67 |
| | Cities <100,000 residents | 1.01 | 0.85,1.19 | 0.94 |
| Delivery in a private hospital | No | Reference | – | – |
| | Yes | 0.86 | 0.82,0.92 | P<0.001 |
| Parental consanguinity | No | Reference | – | – |
| | Yes | 1.19 | 1.12,1.28 | P<0.001 |

*Intrauterine Fetal Demise,

**Venereal Disease Research Laboratory test

 

our results, by using 3 different standards the prevalence of IUGR in the same sample ranged from 4.2 to 11.8%, with one rate almost 3 times higher than the other.

We also studied the potential factors associated with IUGR. According to our findings, the strongest positive associations belonged to trisomy 21 in the neonate, mother's drug addiction or eclampsia/ pre-eclampsia, whereas the strongest protective factors were residence in Paakdasht city, gestational diabetes in the mother and mother's academic education. We couldn't come up with a reason to explain why Paakdasht or Shahryar have significantly lower rates of IUGR than the other cities. Most residents of the cities located near Tehran city (like Robatkarim, Shahryar, Shahre Ghods, etc.) have similar socio-economic conditions. Most of their residents are working-class people whose workplaces are located in Tehran city but reside in nearby towns and commute between the two. The lack of other Iranian studies comparing IUGR rates between cities also adds to our inability to justify the possible causes of this difference. In an Iranian study in 2011, Jahanian et al. reported that maternal weight gain during pregnancy, history of chronic diseases, maternal occupation, and neonate's sex are associated with IUGR [24]. This is in line with our findings about maternal chronic hypertension but neonate's sex didn't show a significant association in our research. The spectrum of factors associated with IUGR in the literature is substantially wide. Mirzaei et al. in another study, reported that a mother's positive history of IUFD, autoimmune or renal diseases, preeclampsia, hyperemesis gravidarum and hypothyroidism and also delivery type are associated with higher IUGR risk [13]. In our study, IUGR was also associated with eclampsia/pre-eclampsia but the mother's history of previous IUFD, autoimmune diseases or hypothyroidism did not reach the level of statistical significance in the final regression model. As mentioned before, due to chronological order, we did not include delivery type in the final model despite showing a significant relationship with IUGR in the univariate analysis. In another study, Sehested et al. reported gestational hypertension, smoking, and placental infarction as IUGR risk factors [25]. The effect of these factors was also seen in another study by Eftekhar et al. He reported that insufficient prenatal care, hookah smoking and high blood pressure during pregnancy, predispose the fetuses to IUGR [22]. Among these factors, we only had the information about mothers' chronic hypertension or eclampsia/pre-eclampsia which showed the same positive association with IUGR. Although the association between mother's smoking and IUGR has been reported in previous articles [26,27], in our study, such an association was only observed in the univariate analysis, but lost its statistical significance in the multivariate analysis. In some studies, parental consanguinity has been reported to be an IUGR risk factor [28], which is also in line with our findings.

Intrauterine growth restriction is linked with increased perinatal morbidity and mortality. It is manageable if caught early in the pregnancy and its risk factors are identified and dealt with as early as possible (ideally before conception). Some factors such as the mother's age or education level at the time of the current pregnancy or this pregnancy being her first, are not really that modifiable at the individual level whereas some others like the mother's addiction can be modified prior to her pregnancy if the mother seeks and has access to good quality prenatal care. When IUGR is confirmed in a fetus, regarding the cause or gestational age of the fetus, different interventions like the use of corticosteroids or magnesium sulfate or earlier delivery and etc. may be advised to reduce the risk of death or IUGR-related early or late complications in the fetus [7].

Although IUGR is still a serious health issue, few recent studies have investigated its prevalence both in Iran and other parts of the world and some have used low birth weight, small for gestational age or IUGR definitions interchangeably [17,22,29]. Using different definitions for IUGR like birth weight less than 2500 g regardless of the gestational age of the newborn, using the 10th, 5th or the 3rd percentiles as indicators and using different charts to reference the 10th percentiles of weight in each gestational age, making comparison of the results almost impossible. Although using the "less than 2500 g birth weight" definition regardless of the neonate's gestational age is more convenient, but using the low-birth-weight definition instead of IUGR is flawed. Many babies that are born prematurely while having normal growth rates for their gestation, may be identified as IUGR cases by this definition. On the other hand, not all term-born babies with less than 2500 g birth weight are suffering from IUGR. For example, in our study, 13.4% of weighing less than 2500 g

term infants were IUGR-free. Also using the "weighing less than the 10th percentile for gestational age" definition, many real cases may actually weigh more than 2500 g (according to all 3 standards used in this study, the 10th percentile birth weights are all above 2500 g after the 37th week of gestation). In our study, 2.1% of term babies with 2500 g birth weight or more, had IUGR, which would be overlooked if the former definition was used. It seems that studies are needed to first choose the most accurate and "practical enough" definition for IUGR and then use this unified definition in all studies performed in different parts of the world making comparison of the rates between the locations or its change through time in the same location possible.

We observed the highest agreement between the populations' birth weight centiles and the World Health Organization's fetal growth charts, however, could not find any similar studies for comparison, again emphasizing the need for such comparisons and a national or global consensus.

## Strength and limitations

The large sample size and inclusion of all live singleton births into our study in a census manner, along with using 3 different standards for IUGR identification are some points of strength in this study. However, having access to data about the mother's height and weight, weight gain during pregnancy or placental information would have added to its strength.

## Conclusion

The prevalence of IUGR in Tehran province differed regarding the standard charts used, ranging from 4.2 to 11.8%. A global unified practical standard for IUGR definition is gravely needed to make comparisons and measuring the effects of future interventions, possible. In our study, the neonate's Down syndrome and maternal factors like addiction, eclampsia/pre-eclampsia, chronic hypertension, history of abortion, Primigravidity, being older than 35 years and parental consanguinity were positively associated with IUGR where the mother's gestational diabetes, higher levels of education, delivering the baby in a private hospital and living in Paakdasht or Shahryar cities showed rather protective associations.

## Author contributions

**Conceptualization:** Narjes Khalili, Arghavan Haj-sheykholeslami.

**Data curation:** Mohammad Heidarzadeh, Abbas Habibelahi, Arghavan Haj-sheykholeslami.

**Formal analysis:** Arghavan Haj-sheykholeslami.

**Investigation:** Mahtab Toulany, Abbas Habibelahi.

**Methodology:** Narjes Khalili.

**Project administration:** Mahtab Toulany.

**Supervision:** Mohammad Heidarzadeh, Abbas Habibelahi.

**Validation:** Mohammad Heidarzadeh, Abbas Habibelahi.

**Writing – original draft:** Mahtab Toulany, Arghavan Haj-sheykholeslami.

**Writing – review & editing:** Mahtab Toulany, Narjes Khalili, Mohammad Heidarzadeh, Abbas Habibelahi, Arghavan Haj-sheykholeslami.

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
