## [Decision Letter · Decision Letter 0]

PONE-D-24-40176Intrauterine Growth Restriction and its associated factors in Tehran, comparing 3 common standards.PLOS ONE

Dear Dr. Haj-sheykholeslami,

Thank you for submitting your manuscript to PLOS ONE. After careful consideration, we feel that it has merit but does not fully meet PLOS ONE’s publication criteria as it currently stands. Therefore, we invite you to submit a revised version of the manuscript that addresses the points raised during the review process.

We look forward to receiving your revised manuscript.

Kind regards,

Surangi Jayakody, MBBS, MSc, MD

Academic Editor

PLOS ONE

**Journal Requirements:**

Reviewers' comments:

Reviewer's Responses to Questions

**Comments to the Author**

1. Is the manuscript technically sound, and do the data support the conclusions?

Reviewer #1: Partly

Reviewer #2: Yes

Reviewer #3: Yes

2. Has the statistical analysis been performed appropriately and rigorously? 

Reviewer #1: No

Reviewer #2: Yes

Reviewer #3: Yes

3. Have the authors made all data underlying the findings in their manuscript fully available?

Reviewer #1: No

Reviewer #2: Yes

Reviewer #3: Yes

4. Is the manuscript presented in an intelligible fashion and written in standard English?

Reviewer #1: No

Reviewer #2: Yes

Reviewer #3: Yes

5. Review Comments to the Author

**Reviewer #1:** 1.The research objectives are unclear and inconsistent between the abstract and the full text.

2. The background section lacks depth and should include: Justification for using three different standards. Reasons for selecting Tehran as the study location, such as its demographic characteristics or relevance to the research context.

3.The term *"evaluate"* does not adequately reflect the research objectives.

4. The study lacks a clear description of the inclusion and exclusion criteria for participants.

5. There is no description of mothers nutrient status ,BMI and anemia levels as potential factors. Is this due to lack of information in the data base ?

6. The discussion section does not specify the standards used in other countries to calculate the prevalence of IUGR. Without this information, comparisons with other studies are unjustified. Include references to the standards used in

few comparative studies.

7. The study does not address its limitations. These should be clearly described, including potential issues related to geographic scope, study population, and generalizability of findings etc.

8. The conclusion is inadequate and should not be the summary of the results. Should highlight the study’s key implications, relevance to practice, and suggestions for future research etc.

9. The article contains numerous grammatical errors. Pls improve the overall quality of the English language.

**Reviewer #2: **Thank you for your great efforts throughout this resaerch article (Intrauterine Growth Restriction (IUGR) in Tehran using three different standards)

It addresses a critical issue in perinatal health and provides valuable insights into the variability in IUGR prevalence based on different growth charts.

However, I have few comments:

Major Concerns:

1- For the definition of IUGR: While you use three common standards (WHO, INTERGROWTH-21st, and local percentiles), the rationale for choosing these over others should be explained more clearly.

2- You state that gestational age was determined based on LMP and ultrasound in irregular cycles. However, were any corrections applied for potential errors in estimation? and why didn't you depend on a dating scan for both groups ( regular and irregular cycles?)

3- Were tests for multicollinearity (e.g., Variance Inflation Factor) conducted? Some variables (e.g., private hospital delivery as a proxy for SES) might be interrelated.

4- The choice of cutoff p < 0.2 for variable selection in the multivariate model is somewhat unfair. Consider referencing established guidelines.

5- The variation in prevalence (4.2% to 11.8%) is a major takeaway. However, the discussion does not fully explore why WHO charts might overestimate or INTERGROWTH-21st might underestimate IUGR cases.

6- You acknowledge the lack of previous Iranian studies using these standards but do not explore potential cultural, genetic, or environmental factors influencing IUGR rates.

Minor Concerns:

The conclusion is vague, and please add recommendations

**Reviewer #3:** Congratulation on reviewing a huge dataset. I read the study with great interest.

The study uses a large, population-based dataset (187,031 singleton live births)- using all 26 weeks onwards births in a census type data collection- a strong dataset for decision making. It describes associated factors on diagnosis of IUGR in the described population using multiple definitions for IUGR.

I agree that the identification of associations for IUGR is very important, especially with regards to planning healthcare delivery and interventions applicable to the population. However I'm much less certain of the external validity of those associations in a different setting.

I have few suggestions on improving the clarity of the current manuscript.

The authors report a stillbirth rate of 0.7 for the study population. Many authors have described stillbirth undercounting in registry based data previously. Eg. Dandona R, George S, Majumder M, Akbar M, Kumar GA. Stillbirth undercount in the sample registration system and national family health survey, India. Bull World Health Organ. 2023 Mar 1;101(3):191-201. doi: 10.2471/BLT.22.288906. Epub 2023 Jan 26. PMID: 36865608; PMCID: PMC9948498. If this fact has been previously looked into in the study population, it is better to discuss the implications in the discussion.

The abstract does not describe how gestational age was determined (e.g., based on ultrasound or last menstrual period), which is critical for accurately defining IUGR. A description on how gestational dating was performed in the study population should be added.

Please discuss further on the adjustments made for confounding factors in the logistic regression analysis (e.g., socioeconomic status, maternal BMI, or access to healthcare).

The variability in IUGR prevalence based on different diagnostic standards- which is previously discussed extensively. Please highlight how this research builds on or diverges from previous studies. The current study methodology is unsuitable to decide which diagnostic criteria to agree on for IUGR diagnosis in the described population.

Lines 101-107; it is unclear of the origin of the denominator (170843).

Discussion- The authors have identified associations with a diagnosis on IUGR in their population. As expected, the prevalence differs depending on the definition of IUGR, ie. the chart used.

Recommendations

In multivariate analysis authors have identified associations with limited biological plausibility for causation- eg residing district, which can be discussed with recommendations for further studies.

Limitations- Please add a statement discussing the study limitations including potential external validity.

6. PLOS authors have the option to publish the peer review history of their article (what does this mean?). If published, this will include your full peer review and any attached files.

Reviewer #1: No

Reviewer #2: No

Reviewer #3: **Yes: **Indu Asanka Jayawardane

---

## [Author Response · Author response to Decision Letter 1]

1 Mar 2025

Dear Dr. Jayakody,

I hope this message finds you well. I would like to express my sincere gratitude for the thoughtful and constructive feedback provided by the reviewers for my manuscript titled “Intrauterine Growth Restriction and its associated factors in Tehran, comparing 3 common standards” (Manuscript ID: PONE-D-24-40176). I appreciate the time and effort they have invested in reviewing my work.

I have carefully considered all the comments and suggestions made by the reviewers, and I am pleased to inform you that I have revised the manuscript accordingly. Below, I provide answers to each of the reviewers’ comments and explain how I have addressed them.

Reviewer 1 Comments:

1. The research objectives are unclear and inconsistent between the abstract and the full text.

I have revised both sections now. I hope they can show that we aimed to measure IUGR prevalence in Tehran, identify the associated factors, and investigate the impact of using different standards for at-birth IUGR recognition on its prevalence in the same population.

2. The background section lacks depth and should include Justification for using three different standards. Reasons for selecting Tehran as the study location, such as its demographic characteristics or relevance to the research context.

The reason behind choosing Tehran province for the study is now provided in the last paragraph of the background. We decided to compare these standards because there was no international or national consensus on using any specific standard. This point is mentioned in the abstract’s objective and also in the manuscript’s background. We tried to show the important impact it can have on IUGR rates and the consequent decisions that can be made on this basis. These 3 standards were selected as they were more frequently used in previous studies, and 2 of them (the World Health Organization charts and the INTERGROWTH-21st charts) claimed to be appropriate for use in any population, and the third one was recommended by the literature for use considering every nation’s specific characteristics.

3. The term *"evaluate"* does not adequately reflect the research objectives.

It is now changed to “assess” both in the abstract and the background sections.

4. The study lacks a clear description of the inclusion and exclusion criteria for participants.

It was mentioned in the second paragraph of the methods section:” All singleton live births with a gestational age of at least 26 weeks in Tehran during 2018 were included in the study in a census manner.” I have now put it under the inclusion criteria title and explained that those with missing data on birth weight or gestational age were excluded.

5. There is no description of mothers' nutrient status, BMI, and anemia levels as potential factors. Is this due to a lack of information in the database?

Yes. We could only use the information gathered in the national registry, which lacks the mother nutrient status and anemia level variables. The BMI variable is included in the system but not among the mandatory information to be reported. This results in many missing values that limit the use of this variable.

6. The discussion section does not specify the standards used in other countries to calculate the prevalence of IUGR. Without this information, comparisons with other studies are unjustified. Include references to the standards used in a few comparative studies.

This is one of the most important messages of this article, emphasizing that the standard we use for IUGR recognition greatly impacts its prevalence, making comparisons between countries or even studying the temporal trend of IUGR prevalence in the same location nearly impossible. Many of the studies referenced in this article did not even mention what standard chart they used to decide which infant had IUGR and only sufficed to say that infants with birth weights less than the 10th percentile were considered to have IUGR. Among the referenced studies just one (reference number 15) had clearly stated what standard they used which I added to the text however for the others (reference numbers 4,5,16 and 17) no specific standard was mentioned, again showing the lack of attention to the great impacts it can have on IUGR prevalence and the grave need for a consensus on which one to use.

7. The study does not address its limitations. These should be clearly described, including potential issues related to geographic scope, study population, generalizability of findings, etc.

Multiple sentences were added to the study’s strengths and limitations section accordingly.

8. The conclusion is inadequate and should not be the summary of the results. Should highlight the study’s key implications, relevance to practice, and suggestions for future research, etc.

A Suggestion for future research is now added to the conclusion. The study’s key implications and relevance to practice is that “a global unified practical standard for IUGR definition is gravely needed to make comparisons and measurement of the effects of future interventions, possible” which was mentioned in the conclusion before.

9. The article contains numerous grammatical errors. Pls improve the overall quality of the English language.

I have used the Grammarly app to improve the quality of the manuscript’s English language. I hope that this time, you find the quality acceptable.

Reviewer #2: Thank you for your great efforts throughout this research article (Intrauterine Growth Restriction (IUGR) in Tehran using three different standards)

It addresses a critical issue in perinatal health and provides valuable insights into the variability in IUGR prevalence based on different growth charts.

However, I have a few comments:

Major Concerns:

1- For the definition of IUGR: While you use three common standards (WHO, INTERGROWTH-21st, and local percentiles), the rationale for choosing these over others should be explained more clearly.

It is now modified accordingly.

2- You state that gestational age was determined based on LMP and ultrasound in irregular cycles. However, were any corrections applied for potential errors in estimation? and why didn't you depend on a dating scan for both groups ( regular and irregular cycles?)

As mentioned in the methods section, we extracted the data from a national registry, and the protocol used for determining the gestational age in that registry is based on LMP in women with regular cycles and ultrasound (preferably performed in the first trimester of the pregnancy) in those with irregular cycles. We have no control over what information is gathered and how it is determined for the registry. I believe that the authorities have chosen this protocol because it can be done all over the country, even in the least developed regions, since it is the cheapest method and needs minimum interventions.

3- Were tests for multicollinearity (e.g., Variance Inflation Factor) conducted? Some variables (e.g., private hospital delivery as a proxy for SES) might be interrelated.

Since our dependent variable was a dichotomous one, we had to use logistic regression, and the assumptions for this test were checked, not the assumptions for linear regression. Also, all of our independent variables were categorical. However, I did use linear regression to calculate the VIF for you but I don’t think that this test is appropriate for these variables. The highest amount for VIF belonged to the mother’s education level with a VIF of 1.45 and tolerance of 0.68.

4- The choice of cutoff p < 0.2 for variable selection in the multivariate model is somewhat unfair. Consider referencing established guidelines.

This Cutoff value is frequently used in medical literature and can be found in the book “Regression Methods in Biostatistics. Authors Eric Vittinghoff, David V. Glidden, Stephen C. Shiboski, Charles E. McCulloch.” Below, you can also find the links to some recent articles that have also used this method:

https://www.ncbi.nlm.nih.gov/pubmed/39167177

https://www.ncbi.nlm.nih.gov/pubmed/38565912

https://www.ncbi.nlm.nih.gov/pubmed/36632097

https://www.ncbi.nlm.nih.gov/pubmed/32153956

5- The variation in prevalence (4.2% to 11.8%) is a major takeaway. However, the discussion does not fully explore why WHO charts might overestimate or INTERGROWTH-21st might underestimate IUGR cases.

To me, that is the most important finding that shows how important it is to have a common uniform standard for use in every article. Many of the relevant articles that I found did not even mention which standard was used to reference the 10th birth-weight percentile and only sufficed to say those weighing less than the 10th percentile at birth were considered IUGR, totally ignoring the fact that the standards can have such a huge impact on the rates. Considering your comment, since we do not have a national or global consensus on which chart gives us the true rate for at-birth IUGR diagnosis, I can’t be certain which of these 3 rates is the real rate of IUGR in this community and decide which one is over or under-diagnosing. Right now, each one of these numbers can be true. I think just showing the big difference between them sends the main message to the audience.

6- You acknowledge the lack of previous Iranian studies using these standards but do not explore potential cultural, genetic, or environmental factors influencing IUGR rates.

Since the Tehran province is a great mixture of different ethnic, socio-economic, and cultural characteristics of people from all around the country, there is no specific characteristic that can be exclusively said about it. It is clear that other provinces with different characteristics, particularly regarding those factors associated with IUGR occurrence, may have different rates. For example, in the least developed regions where mothers have lower levels of education, IUGR may be more common, or in the areas with tribal structure, parental consanguinity may be more common, which can be associated with higher IUGR rates.

Minor Concerns:

The conclusion is vague, and please add recommendations

It is now modified accordingly.

Reviewer #3: Congratulation on reviewing a huge dataset. I read the study with great interest.

The study uses a large, population-based dataset (187,031 singleton live births)- using all 26 weeks onwards births in a census type data collection- a strong dataset for decision making. It describes associated factors on diagnosis of IUGR in the described population using multiple definitions for IUGR.

1-I agree that the identification of associations for IUGR is very important, especially with regards to planning healthcare delivery and interventions applicable to the population. However, I'm much less certain of the external validity of those associations in a different setting.

Thank you for your positive feedback. You are absolutely right; I have added a few sentences about the generalizability of these results in the discussion section under the strengths and limitations title. However, to me, the great difference in rates when using different standard charts is the main finding, which does not depend on the population it's used on or the characteristics of that population.

I have a few suggestions on improving the clarity of the current manuscript.

2-The authors report a stillbirth rate of 0.7 for the study population. Many authors have described stillbirth undercounting in registry based data previously. Eg. Dandona R, George S, Majumder M, Akbar M, Kumar GA. Stillbirth undercount in the sample registration system and national family health survey, India. Bull World Health Organ. 2023 Mar 1;101(3):191-201. doi: 10.2471/BLT.22.288906. Epub 2023 Jan 26. PMID: 36865608; PMCID: PMC9948498. If this fact has been previously looked into in the study population, it is better to discuss the implications in the discussion.

In the extracted data from the national registry, we had one variable that reported if the baby was born alive or not, and we used that to measure the stillbirth rate; of course, that can have deviations from the true rate in the community, but that was the only information that was available to us.

3-The abstract does not describe how gestational age was determined (e.g., based on ultrasound or last menstrual period), which is critical for accurately defining IUGR. A description on how gestational dating was performed in the study population should be added.

It is explained in the methods section of the manuscript’s body, but the word count limit in the abstract didn’t let us to also describe it in the abstract. The gestational age was calculated using the first day of the last period (LMP) in mothers with regular menstrual cycles or the first-trimester ultrasound results in mothers with irregular menstrual cycles or those who could not remember their LMP.

4-Please discuss further on the adjustments made for confounding factors in the logistic regression analysis (e.g., socioeconomic status, maternal BMI, or access to healthcare).

I have added a note under Table 2 that explains that the provided results are adjusted for all other variables present in that table, which includes the mother’s age, level of education, first pregnancy, history of abortion, previous IUFD, chronic Hypertension, autoimmune diseases, hypothyroidism,

gestational Diabetes, Eclampsia/Pre-eclampsia, positive VDRL test, cigarette smoking habit, addiction, alcohol use, city of residence, delivery in a private hospital, parental consanguinity, and

Trisomy 21 in the neonate

5-The variability in IUGR prevalence based on different diagnostic standards- which is previously discussed extensively. Please highlight how this research builds on or diverges from previous studies. The current study methodology is unsuitable to decide which diagnostic criteria to agree on for IUGR diagnosis in the described population.

While this may seem obvious, many studies seem to neglect this issue and even fail to report which standard they used to come up with IUGR rates. Since currently no national or global guideline exists to tell us which chart should be used, this study can first show the huge impact the standards can have on IUGR rates, emphasizing the need for a common practical standard that can be used in all countries. Only when the same standards are used, we can compare the results in different areas or track the temporal changes in IUGR rates in a region.

6-Lines 101-107; it is unclear of the origin of the denominator (170843).

The paragraph is about the term-born infants in our study, categorizing them into those with a birth weight of less than 2500 gr and those weighing 2500 gr or more at birth. There were 174736 term births in our study and among them, 3893 infants weighed less than 2500 gr while the rest of the 170843 neonates had birth weights equal to or more than 2500 gr (3893+170843=174736).

The paragraph shows that while low birth weight and IUGR cases may have some overlaps, they are not the same, and using one instead of the other is incorrect. Unfortunately, in some studies, these concepts are used interchangeably. The results show that although many of our term-born infants with birth weights less than 2500 gr are IUGR cases, a significant proportion of them(13.4%) are not. On the other hand, they show that IUGR diagnosis cannot be ruled out in every term-born infant weighing 2500 gr or more. In our study, IUGR was diagnosed in at least 2.1% of such infants by all 3 charts. This is further explained in the discussion section.

Discussion- The authors have identified associations with a diagnosis on IUGR in their population. As expected, the prevalence differs depending on the definition of IUGR, ie. the chart used.

Recommendations

7-In multivariate analysis authors have identified associations with limited biological plausibility for causation- eg residing district, which can be discussed with recommendations for further studies.

It was corrected. Further research is needed to better investigate the protective effect of living in these 2 cities.

8-Limitations- Please add a statement discussing the study limitations including potential external validity.

It is now

---

## [Decision Letter · Decision Letter 1]

Intrauterine Growth Restriction and its associated factors in Tehran, comparing 3 common standards.

PONE-D-24-40176R1

Dear Dr.Arghavan,

We’re pleased to inform you that your manuscript has been judged scientifically suitable for publication and will be formally accepted for publication once it meets all outstanding technical requirements.

Kind regards,

Surangi Jayakody, MBBS, MSc, MD

Academic Editor

PLOS ONE

Additional Editor Comments (optional):

Reviewers' comments:

Reviewer's Responses to Questions

**Comments to the Author**

1. If the authors have adequately addressed your comments raised in a previous round of review and you feel that this manuscript is now acceptable for publication, you may indicate that here to bypass the “Comments to the Author” section, enter your conflict of interest statement in the “Confidential to Editor” section, and submit your "Accept" recommendation.

Reviewer #1: (No Response)

Reviewer #3: All comments have been addressed

2. Is the manuscript technically sound, and do the data support the conclusions?

Reviewer #1: Yes

Reviewer #3: Yes

3. Has the statistical analysis been performed appropriately and rigorously? 

Reviewer #1: Yes

Reviewer #3: Yes

4. Have the authors made all data underlying the findings in their manuscript fully available?

Reviewer #1: Yes

Reviewer #3: Yes

5. Is the manuscript presented in an intelligible fashion and written in standard English?

Reviewer #1: No

Reviewer #3: Yes

6. Review Comments to the Author

Reviewer #1: Dear , Authors. Most of the concerns raised in the previous manuscript have been answered. But still there are areas to be improved and the manuscript is not adhere to guidelines given for resubmission.

1. Abstract still not reflects the standard quality. Identify the associated factors on what ?, please clarify.

2. Line 81- Please indicate the reference of the text book

3.Line 108-109 is not vey clear. ( fulfilled the criteria of IUGR. Based on which criteria's?)

4..Its still not self explanatory why you have selected only few cities for analysis in Table 01

5. Certain terminologies used in the manuscript is not appropriate for scientific writing. Please modify the manuscript to achieve its standard. (eg. but lost its statistical significance in the multivariate

analysis.../use this unified definition in all studies performed in different parts of the world.....)

Reviewer #3: The suggested comments and amendments have been addressed adequately by the authors. I have no further concerns.

7. PLOS authors have the option to publish the peer review history of their article (what does this mean?). If published, this will include your full peer review and any attached files.

Reviewer #1: No

Reviewer #3: No

---

## [Editor Report · Acceptance letter]

PONE-D-24-40176R1

PLOS ONE

Dear Dr. Haj-sheykholeslami,

I'm pleased to inform you that your manuscript has been deemed suitable for publication in PLOS ONE. Congratulations! Your manuscript is now being handed over to our production team.

Kind regards,

on behalf of

Dr Surangi Jayakody

Academic Editor

PLOS ONE